# Anatomically Precise Microsurgical Resection of a Posterior Fossa Cerebellar Metastasis in an Elderly Patient with Preservation of Venous Outflow, Dentate Nucleus, and Cerebrospinal Fluid Pathways

**DOI:** 10.3390/diagnostics15243131

**Published:** 2025-12-09

**Authors:** Nicolaie Dobrin, Felix-Mircea Brehar, Daniel Costea, Adrian Vasile Dumitru, Alexandru Vlad Ciurea, Octavian Munteanu, Luciana Valentina Munteanu

**Affiliations:** 1Puls Med Association, 051885 Bucharest, Romania; nicolaiedobrin@umfro.com (N.D.);; 2”Nicolae Oblu” Clinical Hospital, 700309 Iasi, Romania; 3Department of Neurosurgery, “Carol Davila” University of Medicine and Pharmacy, 050474 Bucharest, Romania; 4Department of Neurosurgery, “Victor Babes” University of Medicine and Pharmacy, 300041 Timisoara, Romania; 5Department of Pathology, Faculty of Medicine, “Carol Davila” University of Medicine and Pharmacy, 030167 Bucharest, Romania; 6Medical Section, Romanian Academy, 010071 Bucharest, Romania; 7Neurosurgery Department, Sanador Clinical Hospital, 010991 Bucharest, Romania; 8Department of Anatomy, “Carol Davila” University of Medicine and Pharmacy, 050474 Bucharest, Romania

**Keywords:** brain metastases, posterior fossa tumor, cerebellar metastasis, suboccipital craniectomy, elderly brain tumor surgery, dentate nucleus preservation, cerebrospinal fluid pathways, stereotactic radiosurgery, neurosurgical oncology, venous preservation

## Abstract

**Background and Clinical Significance**: Adults suffering from cerebellar metastases are often at high risk for rapid deterioration of their neurological status because the posterior fossa has limited compliance and the location of these metastases are close to the brain stem and important cerebrospinal fluid (CSF) pathways. In this paper, we present a longitudinal, patient-centered report on the history of an elderly individual who suffered from cognitive comorbidities and experienced a sudden loss of function in her cerebellum. Our goal in reporting this case is to provide a comparison between the patient’s pre-operative and post-operative neurological examinations; the imaging studies she had before and after surgery; the surgical techniques utilized during her operation; and the outcome of her post-operative course in a way that will be helpful to other patients who have experienced a similar situation. **Case Presentation**: We report the case of an 80-year-old woman who initially presented with progressive ipsilateral limb-trunk ataxia, impaired smooth pursuit eye movement, and rebound nystagmus, but preserved pyramidal and sensory functions. Her quantitative bedside assessments included some of the components of the Scale for the Assessment and Rating of Ataxia (SARA), and a National Institute of Health Stroke Scale (NIHSS) score of 3. These findings indicated dysfunction of the left neocerebellar hemisphere and possible dentate nucleus involvement. The patient’s magnetic resonance imaging (MRI) results demonstrated an expansive mass with surrounding vasogenic edema and marked compression and narrowing of the exits of the fourth ventricle which placed the patient’s CSF pathways at significant risk of occlusion, while the aqueduct and inlets were patent. She then underwent a left lateral suboccipital craniectomy with controlled arachnoidal CSF release, preservation of venous drainage routes, subpial corticotomy oriented along the lines of the folia, stepwise internal debulking, and careful protection of the cerebellar peduncles and dentate nucleus. Dural reconstruction utilized a watertight pericranial graft to restore the cisternal compartments. Her post-operative intensive care unit (ICU) management emphasized optimal venous outflow, normoventilation, and early mobilization. Histopathology confirmed the presence of metastatic carcinoma, and staging suggested that the most likely source of the primary tumor was the lungs. Immediately post-operation, computed tomography (CT) imaging revealed a smooth resection cavity with open foramina of Magendie and Luschka, intact contours of the brain stem, and no evidence of bleeding or hydrocephalus. The patient’s neurological deficits, including dysmetria, scanning dysarthria, and ataxic gait, improved gradually during the first 48 h post-operatively. Upon discharge, the patient demonstrated an improvement in her limb-kinetic subscore on the International Cooperative Ataxia Rating Scale (ICARS) and demonstrated independent ambulation. At two weeks post-operation, CT imaging revealed decreasing edema and stable cavity size, and the patient’s modified Rankin scale had improved from 3 upon admission to 1. There were no episodes of CSF leakage, wound complications, or new cranial nerve deficits. A transient post-operative psychotic episode that was likely secondary to her underlying Alzheimer’s disease was managed successfully with short-course pharmacotherapy. **Conclusions**: The current case study demonstrates the value of anatomy-based microsurgical planning, preservation of venous and CSF pathways, and targeted peri-operative management to facilitate rapid recovery of function in older adults who suffer from cerebellar metastasis and cognitive comorbidities. The case also demonstrates the importance of early multidisciplinary collaboration to allow for timely initiation of both adjuvant stereotactic radiosurgery and molecularly informed systemic therapy.

## 1. Introduction

Brain metastases remain a prevalent and significant component of systemic malignancy, with an estimated incidence of 8 to 10% in adult cancer patients and an estimated 200,000 new cases annually worldwide. Within this cohort, the cerebellum is involved in approximately 15 to 20% of cases [1]. This frequency demonstrates the cerebellum’s vascular supply and other characteristics that facilitate a metastatic cell’s travel through both the arterial and venous routes. The most common primary malignancies include non-small-cell lung carcinoma, breast carcinoma, melanoma, and colorectal carcinoma; however, virtually any malignant neoplasm with the capacity for hematogenous spread can lead to posterior fossa disease [2,3].

Presentation of cerebellar metastases is characterized by the presence of focal neurological deficits combined with a mass effect in a compartment with limited volume. Manifestations of clinical cerebellar dysfunction including limb and truncal ataxia, dysarthria, nystagmus, and imbalance can develop insidiously; however, it is common for its progression in many patients to be also escalated due to obstructive hydrocephalus or brainstem compression [4]. As any mass produces pressure in a contained environment, even small lesions can manifest as a rapid decline; timely surgical intervention may be required with acute neurological decline, radiological evidence of a mass effect, or when a histopathological diagnosis is needed [5].

The literature details a median overall survival of approximately 9 to 12 months following resection of cerebellar metastases. While survival is influenced by the ability to control systemic disease and pre-operative performance status, post-operative complications may negatively influence survival outcomes [6]. Data from a more recent single-center cohort of patients undergoing resection of cerebellar metastasis report intracranial progression-free survival of 4.5 months; however, a subgroup that had timely adjuvant radiosurgery to the resection cavity had prolonged disease control. These findings highlight the need for integrated multidisciplinary care and timely and appropriate post-operative plans [7].

Current neurosurgical guidance indicates that the plan should be safe resection of posterior fossa metastases, with preferential maximal removal if it is safe to do so; en bloc removal is optimal if the anatomy permits it due to decreased risk of leptomeningeal disease. It is also crucial to avoid compromising venous drainage, avoid traction on the brainstem, and allow for cerebrospinal fluid circulation, each of which affect the patient’s experience post-operatively [8]. Continued advances in pre-operative imaging, functional imaging, and, in certain centers, intra-operative approaches lay the groundwork for a more individualized surgical planning framework, but underlying traditions and principles of microsurgical anatomy still stand unchanged [9].

The developments in oncology hold important implications for the context in which these cases are managed. Increasingly being utilized within the purview of contemporary neuro-oncology, molecular profiling is being studied to determine the risk of developing brain metastases and tumor response to systemic agents [10,11]. Continuing to explore this area, early-phase clinical studies have begun testing the use of targeted therapies and immunomodulators with reported intracranial activity. Imaging advancements—specifically, radiomics-based models—are beginning to enhance the possibility of developing quantitative MRI biomarkers to non-invasively assess resection cavities early on for recurrence or treatment-related changes [12].

In this context, the present case aims to provide a full description of the clinical presentation, radiological features, surgical anatomy, surgical approach, and short-term clinical course of a patient with a significant left neocerebellar metastasis. The aim is to create a clinically relevant documentation for surgical decision-making and intra-operative task-directed execution in future cases, with the goals of anatomical preservation, safe resection, and individualized patient physiology in surgical planning. There are many different aspects of NSCLC brain metastases and an increasing number of options for systemic and radiosurgical treatments for patients; therefore, this report will focus on the anatomy-dependent microsurgical strategy and decisions regarding peri-operative care unique to posterior fossa disease. Although the authors acknowledge a variety of broader topics (i.e., whether or not to perform stereotactic radiosurgery before or after surgery, the use of whole-brain radiotherapy, and various molecularly targeted systemic treatment options) that may provide useful information to surgeons, we have only included them where they specifically impact surgeons’ decision-making. A post-operative MRI performed in a time frame from 48 to 72 h post-operation was not performed due to the fact that the patient had an excellent rapid neurological recovery and transporting the patient at that time would have posed a significant risk to the patient’s cognitive and physiological well-being; the adequacy of decompression and the patency of the patient’s cerebrospinal fluid outlets were confirmed by a non-contrast CT scan, and the patient will undergo a contrast-enhanced MRI during the simulation for her stereotactic radiosurgery when the definition of the cavity will be most relevant.

## 2. Case Presentation

An elderly, 80-year-old female was referred to our neurosurgical unit for urgent assessment of a gradually progressive neurological syndrome that had the time course and characteristics of a space-occupying lesion in the posterior fossa. Her past medical history included Alzheimer’s disease with moderate cognitive impairment, essential hypertension (Stage II) with Stage II hypertensive retinopathy, advanced cervical discopathy (Stage IV), and chronic axial musculoskeletal pain. Her medication history included memantine 10 mg taken daily, a neurotrophic supplement consisting of acetyl-L-carnitine and phosphatidylserine two times a day, haloperidol (5 drops) three times a day, trazodone in a fractional evening regime equal to 150 mg, antihypertensive therapy (esp. involving diuretics) titrated to systolic blood pressure between 120 and 140 mmHg, and paracetamol 500–1000 mg as needed. She denied tobacco, alcohol, or illicit drug use, and had no known drug allergies.

The patient was functionally independent in ambulation, and had not used gait aids despite comorbidities up until approximately 3 months previously, when her family noticed a slight increase in the base of her gait, occasional left foot outward deviation during the swing phase, and slight leftward drift during walking. Over that period, her gait disturbances became more constant, and she increasingly relied on environmental contact for balance; hesitated when initiating gait from a sitting position; and noted truncal sway during quiet stance. There was no associated vertigo, nausea, or headache. Her cognitive profile deteriorated in parallel with disturbances in gait. For example, she was observed to have delays in processing multi-step commands; there were increases in the time taken to name a single object, and she experienced random moments of spatial disorientation in an environment that was otherwise familiar. The changes in her cognitive performance earlier in this paragraph fluctuated with respect to severity throughout the day and suggested the possibility of intermittent perilesional edema or perfusion variation. At the time of admission, she was calm, fully cooperative, fully participating, orientated to self and place, and only mildly disoriented to the date. Her scores were 22/30 (MMSE) and 18/30 (MoCA); her MMSE score was usual for her baseline trajectory of Alzheimer’s with a further modest decline. Her impairments on the MMSE were in visuo-spatial/executive function and delayed recall. Her NIH Stroke Scale (NIHSS) score was 3 (points given for slight dysarthria and limb ataxia). Additionally, on review, her total score on the Scale for the Assessment and Rating of Ataxia (SARA) was 13/40, and her subscores are available for gait, stance, sitting, speech disturbance, finger chase (left), nose–finger test (left), fast alternating hand movements (left), heel–shin slide (left). Her gait subscore was 3/8. Her stance subscore was 3/6. Her sitting subscore was 0/4. Her speech disturbance subscore was 1/6. Her finger chase left subscore was 2/4. Her nose–finger left subscore was 2/4. Her fast alternating hand movements left subscore was 1/4. Her heel–shin slide left subscore was 1/4. Her total score on the Tinetti Performance-Oriented Mobility Assessment was 14/28, indicating high risk of fall.

Her speech was clear and articulate, with appropriate modulation of voice volume; however, she engaged indecisively in the phonetic analysis as it was a laborious process where she exhibited some scanning dysarthria, assessed as irregularity of her syllabic rhythm, and to some degree even showed prosodic flattening. In her disjointed “pa-ta-ka” reproductions of the rapid syllable sequences, the left side outputs had key speech elements slowed after 6–7 syllables, which resulted in a lack of consistent intervals, lack of precision and consistency of articulation, and subtle vowel distortion. For short sentences, we rated her accuracy of articulation to be around 95%, with an estimated accuracy of about 88% for longer utterances (polysyllabic). Upon cranial nerve examination, she demonstrated full visual acuity and visual fields (although she was slow to respond when using high-contrast stimulus; in the left inferior quadrant, her response time was 280 ms on average over 5 trials). The pupils were equally and briskly reactive; no relative afferent pupillary defect was recognized. Fundoscopy examination showed sharp optic disk margins and no evidence of optic disk pallor/edema. Smooth pursuit to the left demonstrated a gain of 0.82 (0.96 to the right), and smooth pursuit was interrupted by saccades; the target onset of saccades to the left was delayed ~210 ms compared to right saccades in repeated trials. Gaze was maintained in the eccentric left position for a long period (≥20 s) and resulted in 3–4 beats of rebound nystagmus. Optokinetic nystagmus verified that the amplitude of leftward response was symmetric, although some subtle narrowing of slow-phase velocity occurred to the left for the leftward target. The near point of convergence was normal to 35 cm. She had a bilaterally intact vestibulo-ocular reflex. Sensation of the face was symmetric on all branches, and there was no difference comparing left or right supraorbital regions in the lag of the blink reflex (44 vs. 37 ms average). Her face was symmetric, although her left eyebrow elevation (goniometer) was 8–10% less than the right side. Hearing was symmetric with the whispered voice and 512 Hz tuning fork. Palatal elevation was midline, there was no difference in the gag reflex, and the tongue protruded symmetrically with no fasciculations.

The motor assessment confirmed normal bulk and muscle tone. Her strength was 5/5 (MRC) bilaterally, and there was no evidence of pronator drift. There was a 20% decrement in sustained abduction (isometric), with the left shoulder with the handheld dynamometer at 15 s, but there was full recovery after (or change in contraction). There was a mild similar left elbow flexion deficit (12% decrement). Coordination testing found left-sided dysmetria on the finger-to-nose test, with endpoint tremor amplitude of 1.5 cm, with consistent overshoot The right side was intact. The left heel-to-shin motion was decomposed, slowed, and was completed 5.6 s slower than the right. Left rapid alternating pronation–supination hand movements degraded in amplitude and frequency after 6 cycles, with a 22% reduction in cycle rate; the right hand was stable. Left heel tapping rhythm was lost after 10 taps; inter-tap intervals increased from 0.5 s to 0.8 s. Left finger chase testing produced average initiation delays of 320 ms and an overshoot of 1–1.2 cm for most trials. The rebound phenomenon was positive on the left and absent on the right. Station and gait testing indicated the patient was unable to sustain tandem gait for more than 2 steps without lateral sway. Regarding Romberg testing, sway amplitude was approximately ~3 cm with eyes open and approximately ~9 cm with eyes closed, with examiner support. The gait was broad-based and stance duration prolonged on the left (68% of gait cycle) and the contralateral swing phase was shortened. Turning was slow and required 4–5 corrective steps to regain stability. Sensory testing including light touch, pinprick, vibration sense, and joint position sense, which were normal throughout. Vibration sense decay at the great toe was 13 s bilaterally; for two-point discrimination at the index finger, the left was 3.8 mm and the right was 3.1 mm. Graphesthesia and stereognosis were intact bilaterally. The diminished deep tendon reflexes were brisk and symmetric; plantar responses were flexor bilaterally.

This constellation of findings—strictly ipsilateral limb and truncal ataxia, scanning dysarthria, rebound nystagmus, impaired smooth pursuit with reduced gain, saccadic initiation delay, prolonged blink reflex latency, quantifiable fatigability with sustained isometric contraction, and pyramidal and sensory modalities preserved—anatomically localized the lesion to the left neocerebellar hemisphere, with probable involvement of the dentate nucleus and associated efferent dentatothalamocortical pathway, with medial depression to the vermis involving vestibulocerebellar projections. Further, in the context of a progressive course, with a limited volume to compensate in the posterior fossa and preserved distal reserve (ASA Physical Status III, Clinical Frailty Scale 4), urgent high-resolution neuroimaging was warranted to define the lesion size and structural characteristics and the surgical approach prior to reflex neurological decompensation.

Magnetic resonance imaging of the brain (Figure 1) was performed using multiplanar acquisitions in T1-weighted, T2-weighted, fluid-attenuated inversion recovery (FLAIR), and susceptibility-weighted sequences. The study revealed a single space-occupying lesion in the left cerebellar hemisphere with morphology and displacement that mirrored the patient’s neurological examination.

On the coronal T1-weighted imaging (Figure 1A), the lesion was associated with iso- to mildly hypointense signal compared with neighboring cerebellar cortex, with a curved convex lateral border extending the hemispheric contour. The superior border followed the tentorial surface with small intervening CSF, and the medial contour extended into the superior vermis, where it caused a subtle flattening of the midline contour. Its inferior pole extended to slightly above the level of the cerebellar tonsil and had a small amount of CSF within the cisterna magna. The mass effect was generating medial displacement of the fourth ventricle to the right and decreasing the caliber of the left lateral recess. The mass effect contended for some space to partially narrow the foramen of Luschka without cranial dilation of the fourth ventricle, as is typically observed with significant upstream ventricular dilation. Aside from the lateral recess being displaced midway, there was a small rotation of the ventricular axis, a characteristic that is clinically relevant to intra-operative navigation in this confined area of the posterior fossa. On the coronal T2-weighted image (Figure 1B), the lesion was seen as significantly hyperintense with internal signal heterogeneity, suggesting variability in tissue components as well as localized fluid. There was a large T2 signal in the surrounding cerebellar white matter which extended medially into the paravermian white matter and anteriorly toward the middle cerebellar peduncles, which suggested there was diffuse vasogenic edema. The edema assumed a pathway that probably followed the dentatothalamic efferents traversing the superior cerebellar peduncle and the afferent corticopontocerebellar fibers passing through the middle cerebellar peduncle, both of which are functionally important tracts. The superior margin of the lesion created a subtle elevation and flattening of the tentorial insertion line on the left, indicating transmitted mass effects into the supracerebellar compartment. The coronal FLAIR sequence (Figure 1C) provided excellent definition to the margins of the lesion and edema, as the signal from CSF was suppressed. The hyperintense halo of edema extended from the lateral hemisphere inwards to the paravermian area, narrowing the perivermian cistern and partially effacing the posterior recesses under the fourth ventricle. The dorsal surface of the pons appeared to be slightly flattened and displaced anteriorly, but the aqueductal inlet still appeared to be patent. The angular formation of edema, which was directed medially and anteromedially, suggested that the primary vector of pressure from the lesion was pressing up against the midline vermian structures and the brainstem–cerebellar interface rather than being directed inferiorly towards the foramen magnum. Both the coronal and sagittal FLAIR (Figure 1D) sequences provide a complete anteroposterior view of the lesion spanning anteriorly near the posterior medullary velum to the convex posterior cortical surface of the cerebellar hemisphere. The upper margin was very slightly apposed to the lower surface of the tentorial leaflet in a position that was physiologically compressed such that there was a thin plane of CSF interfacing the midbrain and the tentorial tentorium layer, limiting the potential upward cerebellar ‘relaxation’ during the exposure. The inferior interface was likely approximately 6 mm from the cranial foramen magnum with some preservation of the cisterna magna and its ancillary role in controlling CSF release at the initial opening of the dura. The dorsal displacement of the brainstem was exaggerated in this imaging plane. Note that there was also gentle anterior bowing of the superior medullary velum. The axial susceptibility-weighted image (Figure 1E) showed multiple punctate hypointense foci in the tumor, likely representing distant intralesional blood product deposition, either hemosiderin or deoxyhemoglobin, and presenting some evidence of a lesion with vascular channels supporting microhemorrhage. The sharp interface noted between the lesion core and surrounding hyperintense edema, and the absence of satellite nodules or evidence of multifocal disease within the posterior fossa, again suggested a well-defined lesion. While the cranially located vein of cerebellopontine fissure and the adjacent cortical veins from the hemispheric were open, the proximity of the adjacent draining veins highlighted the need to protect venous drainage during tumor exposure and resection.

In sum, the imaging described a tumor that was well-circumscribed and expansile in the left neocerebellar hemisphere where there was a mass effect toward the vermis medially, to the brainstem anteriorly, and toward the tentatorial insertion anteriorly. However, based on the immediate degree and direction of displacement with imaging and proximity to the fourth ventricle and outlets, even the potential for obstructive hydrocephalus remained precarious, such that even small further lesion enlargement or intratumoral hemorrhage could lead to acute obstruction. The proximity of the tumor to the middle cerebellar peduncle and dentate nucleus illustrated that operative dissection would need to take account of both the afferent and efferent connections of the cerebellum and the deep cerebellar nuclear microarchitecture identified within the tumor. The vascular signature demonstrated in the SWI suggested that if intratumoral decompression were attempted, then intratumoral decomposition would need to be undertaken with stepwise hemostasis with early control of arterial feeders and selective attention to preserve the integrity of venous drainage. As this pre-operative imaging established the anatomical and radiological relationships, it may have ultimately guided and directed the surgical approach, including working angles and dural opening, and consideration of the later order of tumor debulking/capsule dissection.

The patient was positioned in a right park-bench position under general anesthetic, with the left cerebellar hemisphere elevated above the operating table. The head was secured to a Mayfield clamp using three pins, then tilted downward at approximately 30° and forward at about 12° in order to orient the operative trajectory perpendicular to the largest dimension of the tumor, as depicted by pre-operative MRI. By positioning the posterior aspect of the tumor first (the point of entry), the surgeon’s hands (operating from the patient’s left side) would be optimally positioned to dissect within the deep lateral hemisphere of the cerebellum. To prevent elevation of posterior fossa pressure during the most delicate aspects of the dissection process, the neck was adjusted to maximize venous outflow through the jugular veins using intermittent hand-held surface Doppler probes over both jugular veins following every incremental adjustment of the Mayfield clamp. Assessment of waveform amplitude and phasicity were spot-checked between adjustments of the Mayfield clamp; if there was a decrease in amplitude of the audible signal or absence of expected respiratory variation, minor adjustments to the head position were made until it was symmetric; robust and steady flow was achieved in both jugular veins. Decrease in amplitude below baseline served as the criterion for repositioning to maintain optimal venous drainage throughout the case.

Lateral suboccipital craniectomy was intentionally chosen as opposed to craniotomy because it was the best option for safely and physiologically managing the pressure dynamics within the posterior fossa. The depth of the lesion within the lateral aspect of the cerebellar hemisphere, the close proximity to the dentate nucleus, and anticipated post-operative hyperemia in the elderly brain (which is known to have decreased compliance) resulted in a significant risk of early secondary swelling in this particular case. Using a rigid bone flap would provide little buffer space for expansion and could potentially transmit pressure onto the newly constructed cisternal spaces, which could be catastrophic; additionally, any compromise of venous outflow from the posterior fossa—which was already the primary force behind intracranial turgor—posed an excessively high risk of brainstem deformation, delayed awakening, or acute post-operative neurologic deterioration. Although a craniotomy was technically possible, theoretically there are no clinically relevant benefits to a craniotomy: even though the cosmetic appearance of the scalp could be completely covered by the suboccipital musculature and the stability of the musculoligamentous sheath could be maintained regardless of whether bone was replaced, the potential for compression of the sinuses or impairment of drainage of post-operative edema presented a clear and avoidable hazard. Therefore, the decision to leave the bone off was the obvious choice as the safer and more reliable approach. The duralplasty was performed in a watertight manner using a pericranial graft and reinforced with Valsalva leak testing until no efflux occurred, and each mastoid air cell was meticulously filled with bone wax to ensure that there were no pathways for post-operative cerebrospinal fluid leakage.

Upon dural exposure, the tautness of the dura confirmed elevated intracranial pressure within the posterior fossa compartment. A C-shaped dural incision was made with its base toward the transverse and sigmoid sinuses. The flap was reflected anterolaterally and tacked under constant irrigation to prevent cortical desiccation. The cerebellar hemisphere appeared mildly tense and congested, with superficial bridging veins visible through the pial surface. The arachnoid over the cerebellomedullary cistern was sharply opened, releasing clear CSF from the foramen of Magendie. With each 2–3 mL release, the hemisphere visibly softened. Additional arachnoid fenestration over the cerebellopontine cistern further reduced turgor, revealing the petrosal surface and several hemispheric veins. The superior petrosal vein complex was identified early, its tributaries mapping cortical territories of the superolateral hemisphere. Each was preserved by working within the fine arachnoid veil rather than directly mobilizing their walls. Preservation of these venous pathways was critical to preventing delayed cerebellar swelling and venous infarction. Microsurgical exploration revealed the tumor capsule 25 mm deep to the cortical surface, approached through a 15 mm subpial corticotomy placed in line with folial orientation to preserve transverse fiber integrity. The capsule was firm, pale gray, and lobulated, with a thin arachnoid layer separating it from adjacent gliotic cortex. Pial vessels crossing the capsule were mobilized by elevating the arachnoid sheet from the capsule, preserving their course to the cerebellar cortex.

Arterial feeders from hemispheric branches of the superior cerebellar artery (SCA) and anterior inferior cerebellar artery (AICA) were encountered at the capsule surface. Each was carefully dissected free of tumor tissue, coagulated with pinpoint bipolar application at low thermal load, and divided, leaving perforators to the dentate nucleus and brainstem untouched. Internal debulking began with piecemeal aspiration of the tumor core using suction and fine bipolar forceps, working centrifugally to allow the capsule to collapse inward. Tumor removal followed a precise sequence:Posterior pole decompression created space for inward collapse.Lateral wall mobilization allowed the hemisphere to fall medially under gravity, widening the operative corridor without retractor use.Superior surface dissection along the tentorial undersurface preserved bridging veins to the tentorial sinus.Medial surface separation from the paravermian zone and dentate nucleus was performed millimeter-by-millimeter, using cottonoid countertraction and sharp dissection to preserve dentatothalamocortical fibers.Anteromedial pole detachment from the middle cerebellar peduncle was conducted with microdissectors alone, avoiding bipolar contact with the densely packed pontocerebellar fibers.

Throughout the procedure, the tactile feedback from the gliotic plane was critical: firm resistance indicated functional parenchyma, whereas friable adherence suggested tumor infiltration. Irrigation maintained optical clarity and tissue hydration, and minor venous oozing was controlled with oxidized cellulose applied without compression that could kink preserved veins. The final resection cavity showed the following:Smooth medial wall, following the vermian contour without midline violation.Superior wall in continuity with tentorial dura, bridging veins intact.Anterior wall respecting the middle cerebellar peduncle contour.Inferior wall terminating above the foramen magnum, preserving cisterna magna.

Hemostasis was meticulous and selective, avoiding indiscriminate coagulation. The arachnoid was loosely re-approximated over the cavity to restore CSF compartmentalization. Dural closure was watertight with a pericranial graft secured 3 mm from sinus margins. Bone edges were smoothed, mastoid air cells re-sealed, musculofascial layers reconstructed anatomically, and the skin closed in layers. The patient was transferred, intubated, to the neurosurgical intensive care unit (ICU) for elective overnight monitoring, with the head of her bed 30° elevated to favor passive venous return from the posterior fossa. Normoventilation (PaCO_2_ 35–38 mmHg) and normoxia (PaO_2_ > 100 mmHg) were well maintained throughout the immediate post-operative period, and care was taken to avoid rapid fluctuations in arterial blood pressure that could negatively influence cerebellar venous drainage, which was already compromised. Core temperature was maintained within physiological limits in order to augment the metabolic insult on perilesional tissue.

A non-contrast cranial computed tomography (CT) scan performed within two hours of dural closure (Figure 2) demonstrated not only complete evacuation of the space-occupying tumor but also that the left cerebellar hemisphere was decompressed. At the inferior cerebellar convexity (A) of the axial sections, the resection cavity occupied the expected shape of the left neocerebellar hemisphere posteriorly, and the margin of the craniectomy constituted the right lateral wall. There was no residual mass effect on the foramen magnum. More rostrally (B) in the axial images, the cavity shape remained foramellous and adhered to the superior cerebellar surface under the tentorium with the bridging veins intact, and without hyperdense foci, suggesting venous thrombosis. At the level of the fourth ventricle (C), the shape of the midline vermis was intact, and the ventricle outlet pathways (foramina of Luschka and Magendie) were patent so that CSF could cross these barriers unrestricted to cisterna magna. No intraventricular or parenchymal hemorrhage was detected, and pericavitary hypodensity consistent with mild post-operative edema was <4 mm thick. The brainstem contours, particularly the dorsal pons and middle cerebellar peduncle, were crisp, undistorted, and undelimited. Although a post-operative gadolinium-enhanced MRI (early within 48–72 h after surgery) is often helpful in demonstrating the post-operative resection cavity and providing definition of surgical margins for stereotactic radiosurgery planning, no such image was obtained in this case. This patient stabilized neurologically and continued to improve in an uneventful manner with no evidence from either imaging studies or clinical findings to necessitate an urgent reassessment using intravenous contrast. Due to the patient’s advanced age and pre-existing cognitive impairments, the care team felt that avoiding the initial post-operative transportation and potentially sedating the patient would help to prevent additional physiologic stresses to the patient.

Standard intra-operative posterior fossa neuromonitoring was employed to monitor the brainstem and cranial nerves; this included the use of brainstem auditory evoked potential (BAEP), and cranial nerve electromyography of CN V, VII, IX, X, and XI, in addition to somatosensory (SEP) and motor-evoked potentials (MEP). A stable baseline signal existed, and no major amplitude depression or latency shift was seen during the entire surgical procedure; the symmetry and reproducibility of the BAEP waveform existed at time of closure. Osmotherapy was not deemed necessary due to the patient’s age and hemodynamic stability. Antibiotic prophylaxis was provided via a single pre-incision dose of cefazolin. Dexamethasone (8 mg q 8 h for the first 24 h post-op) was continued from the pre-op period and tapered out in 5 days as clinical evidence of edema decreased. Low levels of analgesic requirement were noted post-op; scheduled acetaminophen was utilized with infrequent administration of low-dose opioid for rescue analgesia on the initial post-operative evening.

After awaking from anesthesia, the patient was alert, fully oriented, and could follow a series of commands without delay. Pupillary size and reactivity were symmetrical, extraocular movements were full, and smooth pursuit showed near-normal gain on both sides. Skew deviation was not elicited nor was gaze-holding nystagmus. Facial weakness was not present with House–Brackmann grade I function. Speech was clear with mild residual scanning dysarthria, which resolved within 48 h. Banked bedside swallow testing demonstrated an intact gag reflex and no aspiration on a 5 mL swallow of water confirming intact cranial nerves IX and X. Cerebellar testing showed mild left limb dysmetria on finger-to-nose and heel-knee-shin tasks (ICARS limb-kinetic subscore 4/52), but there was no rebound phenomenon. Rapid alternating movements with the left hand were slightly slow compared to the right but were not decomposed. Axial stability was preserved; the patient was able to sit independently immediately post-operation and stand with minimal assistance by post-operative day (POD) 1. All four extremities were tested with full MRC 5/5 strength, deep tendon reflexes were symmetrical, and plantar responses were flexor bilaterally. Intra-operative dural tensitional ICP assessment and post-operative clinical surrogates confirmed overall ICP was within normal limits; no clinical or radiological signs of acute hydrocephalus were noted. Analgesic needs were minimal (a scheduled doses of paracetamol and low-dose opioid (rescue) for the first 24 h). After hemostatic stability was confirmed by repeat CT, prophylactic low molecular weight heparin commenced POD 2. Early mobilization started POD 1 with assisted ambulation advancing to independent ambulation by POD 3. No CSF leak, wound dehiscence, or infections of the surgical sites were evident. The patient remained afebrile with normal inflammatory markers. Repeated neurological assessments demonstrated continued improvement in fine motor coordination with the ICARS limb-kinetic subscore at discharge 1/52. Importantly, no clinical signs of posterior fossa syndrome, brainstem cranial nerve palsy, or delayed cerebellar swelling were evidenced.

The trifecta of complete removal of lesions, preserved venous anatomy, conscientious dural closure, and early re-establishment of CSF physiological dynamics helped to facilitate an uneventful recovery. The patient was discharged home on POD 7 neurologically intact other than mild residual limb-kinetic dysmetria, with outpatient neurorehabilitation planned.

Two weeks post-surgery, the patient showed ongoing neurological recovery and no new deficits. The limb ataxia previously observed on the operated side was nearly gone, only apparent with high amplitude performative or rapidly alternating movements; almost complete endpoint precision with no overshoot was noted with the finger–nose test. Intention tremor was minimal, low amplitude, and non-disabling, occurring only in the final 5–10° of elbow extension. Performance on the heel-to-shin test was smooth and continuous and stability at the trunk was improved, with the trunk no longer swaying while sitting unsupported. The features of the patient’s gait revealed normalizing stride width and a symmetric step cadence, with both tandem walking tests passing with 10 correct steps without the need for correction. The features of speech prosody and articulation demonstrated that range had returned to normalcy, while the quality of smooth pursuit eye movements appeared symmetric without corrective saccades. Cranial nerve function remained intact with full motor and sensory systems intact.

Neuroimaging at this time (Figure 3) demonstrated structural correlates to favorable clinical recovery. The post-operative cavity within the left neocerebellar hemisphere was well shown with axial CT imaging (Figure 3A), with complete collapse of volume of residuum from the previous surgery, with evidence of a mass effect on the fourth ventricle from before the previous surgery. The cavity was well delineated with smooth gliotic cavity walls, with the absence of irregular nodularity implying evidence of residuum. The attenuation of the parenchyma surrounding the cavity appeared normal. The cavity measured superiorly from the superior folial tiers to above the foramen magnum and spared the vermis and dentate nucleus medially while confirming that the undersurface of the tentorium maintained its anatomical relationship without upward displacement. The basal cisterns remained patent, including the prepontine and cerebellopontine cisterns. No signs of hydrocephalus, transependymal CSF seepage, or hemorrhagic sequelae were observed.

The post-operative course within this time interval was uneventful with no surgical site infection, CSF leak, pseudomeningocele, or compromise to wound healing. The functional evaluation was consistent with clinical improvement with the modified Rankin Scale improving from 3 on admission to 1 at this time point. This indicated a highly favorable neurological recovery, with only negligible residual findings detectable based on high-demand coordination testing.

This case is meant to provide a comprehensive account of the clinical presentation, neuroimaging profile, surgical anatomy and short-term post-operative evolution of a patient with a large left neocerebellar hemisphere tumor. The aim of this report is to demonstrate how the combination of high-definition clinical examination, anatomically guided microsurgical planning, and careful intra-operative implementation obtain optimal outcomes, and to emphasize the importance of thorough pre-operative assessment of restrictive anatomy and preservation of functional pathways and their impact on post-operative outcome in posterior fossa surgery. Our goal is to provide a major reference that may be helpful to other clinicians in planning and undertaking similar interventions with complex neurovascular anatomy. While the outcome for this individual was good, the case reminded us that these types of lesions, when possible, need to be approached differently in an anatomy-driven capacity balancing resection with function.

## 3. Discussion

This case report describes an 80-year-old woman who had a singular, contrast-enhancing mass of the left neocerebellar hemisphere. The only clinical manifestation, ipsilateral limb–truncal ataxia, with oculomotor dysfunction and intact long tracts, anatomically localized the lesion to lateral cerebellar networks. Surgical excision, using a left lateral suboccipital approach, obtained immediate decompression and rapid neurologic recovery. Histopathology diagnosed a secondary (metastatic) lesion with indeterminate primary via routine stains; immunohistochemistry was recommended; and staging CT demonstrated a 3 × 2 cm lesion in left lower-lobe lung, suggesting a working diagnosis of cerebellar metastasis primarily from the lung. The post-op period was otherwise unremarkable, other than a brief and remitting psychotic episode as a result of Alzheimer’s; and she was discharged on memantine, haloperidol and trazodone with plans for outpatient oncological follow-up. An en bloc removal was not possible due to the location and relationship to surrounding structures of this tumor. Imaging studies revealed that the medial and inferior edges of the tumor were in contact with the dentate nucleus and followed the course of the superior and middle cerebellar peduncles and left a thin layer of parenchyma between them. The upper pole of the tumor was positioned under the tentorium (the membrane separating the cerebrum from the posterior fossa) and there were limited and very small bridging veins that prevented safe mobilization and made intra-operative traction dangerous. Therefore, a controlled internal debulking was performed to decompress the cerebellar hemisphere while preserving its venous outflow and all important white matter tracts. This approach resulted in a clean, stable cavity which could be clearly outlined at a later date to allow for precise contouring of the cavity wall and all of the surrounding cerebellar tissue for margin planning for a stereotactic radiosurgical treatment.

### 3.1. Epidemiology, Infratentorial Predilection, and Clinical Risks

In adults with brain metastases, 15–25% arise in the posterior fossa, a location prone to unique hazards: small infratentorial compartments, infrequent compliance reserve, and proximity to the fourth-ventricular CSF paths all confer risks for sudden neurological decline, obstructive hydrocephalus, and brainstem compression [13]. The anatomic peculiarities of the infratentorial space often favor interventions that can rapidly ameliorate mass effect. Recent multi-institutional series have further recognized that infratentorial location independently associates with peri-operative challenges and with patterns of leptomeningeal and ependymal failure if adjuvant therapy is appropriately optimized [14]. To situate this case in the developing evidence base, clinical series and early-phase trials have been approached with consideration of what will happen with these specific issues and with potential management options that characterize posterior fossa brain metastases [15]. The issues and options will be different depending on how the series or trial were designed and conducted, but together they provide a mixture of evidence that demonstrates how important tumor biology, lesion anatomy, patient comorbidity. and the timing of adjuvant therapies are and how they all contribute to functional recovery as well as long-term disease control. Table 1 is a summary of selected original studies completed from 2020 to 2025, which are useful for our current treatment decisions.

### 3.2. Patient-Specific Indication for Resection

The indication for microsurgical excision in this patient was guided by the following: (1) there was a solitary lesion and it was resectable using a suitable pial plane; (2) the patient had a progressive cerebellar syndrome with questionable posterior fossa reserve; and (3) there was an opportunity to obtain tissue for a more complete diagnostic work-up (IHC and if indicated, NGS) given the less than definitive histology on scheduled hematoxylin–eosin.

These characteristics are consistent with present decision models that support resection for solitary symptomatic metastases in eloquent infratentorial sites to relieve mass effects and allow for timely adjuvant radiotherapy.

### 3.3. Adjuvant Radiation: Cavity SRS, Timing, and Pre- vs. Post-Operative Sequence

Cavity-directed stereotactic radiosurgery (SRS) is now regarded as the preferred adjuvant option after the gross-total resection of a solitary metastasis, achieving high rates of local control and limiting neurocognitive expense related to whole-brain radiation therapy (WBRT) [24]. Real-world post-operative SRS cohorts in 2024 showed excellent 12-month local control while using individualized dosing (typically 18–20 Gy single-fraction for small-moderate cavities) and careful contouring along the surgical tract. Note that the posterior fossa location presents additional challenges: key aspects to include are the fourth-ventricular outflow and brainstem constraints [25]. The patient’s planned post-operative adjuvant treatment will be conducted according to current standard practice of posterior fossa SRS: an 18 Gy dose given in one fraction is acceptable for cavities of this size; however, if the final simulation demonstrates a portion of the tumor bed has been left near the brainstem and thus requires dose fractionation to respect local tissue tolerance, then a three fraction dose of 21 Gy can be used. This treatment should be initiated between two to four weeks after surgery when post-operative stabilization and cavity delineation have been achieved. The surgical tract will be included as part of the planning target volume. Therefore, the planning contour should extend from the superficial dural entry point to the most caudal extent of the surgical resection plane to minimize the risk of seeding the surgical tract with cancer cells. The design of the posterior fossa plan dictates that the D0.03 cc of the brainstem should receive ≤12.5 Gy (for single-fraction plans) or ≤23 Gy in 3 fractions, and the floor of the fourth ventricle and surrounding cerebellar peduncles are to be spared to allow for a steeper dose gradient while still achieving 100% coverage of the cavity wall. The above constraints represent the optimal balance of a therapeutic window to control the cavity effectively while safely accommodating the compact anatomy of the brainstem–fourth ventricle complex. Figure 4 provides a visual summary of the patient’s presentation, surgical management, post-operative course, and scheduled cavity SRS.

Two complementary advancements are important when counseling patients regarding sequencing. First, the feasibility and early-efficacy outcomes of pre-operative SRS (irradiating the intact lesion 24–72 h prior to surgery) are steadily accumulating; notably, a 2025 prospective phase-2 trial (PREOP-1) demonstrated practical workflow and encouraging cavity control, encapsulated by the biologic rationale of sterilization of tumor-CSF interfaces ahead of any dural opening [22,26]. A randomized trial that compares pre- vs. post-operative SRS is ongoing to determine different risks of leptomeningeal dissemination (LMD) and distant failure. Pediatric patients teach us especially about brain tumor dissemination, including LMD [27]. Second, a systematic analysis (2025) of SRS vs. WBRT established overall survival was roughly equivalent; however, there are indications of LMD after SRS in some contexts—an observation that puts a premium on meticulous coverage of the cavity (including the surgical tract area) and off targeting early (generally within 2–4 weeks) post-resection from posterior fossa [28]. In our patient, early post-operative imaging confirmed decompression, patent CSF outlets, and no evidence of hemorrhage (Figure 2 and Figure 3 in the case presentation), which provided a regulatory substrate for timely adjuvant cavity SRS should consensus be achieved as a component of multidisciplinary care.

### 3.4. Systemic Therapy with CNS Activity: The Implications for Lung-Primary Suspicion

When thorax imaging demonstrates lung primary origin, systemic therapies that are stratified molecularly often impact intracranial control:

EGFR-mutant NSCLC—Osimertinib continues to be the CNS-active first-line standard; longitudinal cohorts will characterize the evolving patterns of brain involvement in 2024–2025, with improvements in intracranial responses and delayed CNS progression compared to earlier generation TKIs [29,30].

ALK-rearranged NSCLC—third generation lorlatinib has provided robust intracranial PFS and reduction in the development of new brain metastases and supports integrated local-systemic strategy sometimes deferring WBRT [31].

While definitive histotype and driver status were not yet available in this case, the recommendation for IHC from the pathology report is in keeping with contemporary diagnostic algorithms for unknown primaries of metastasis; IHC (and, where possible, with DNA/RNA profiling) allows for better attribution of place-of-origin, and in rare cases explicitly unlocks targeted approaches with brain central nervous system activity [32].

### 3.5. Peri-Operative and Geriatric Considerations: Frailty, Cognition, and Delirium

Older age, baseline cognitive impairment, and posterior fossa disease raise the high stakes of peri-operative management. Contemporary geriatric-neurosurgery cohorts (≥65 y) suggest that for older adults, deterioration in baseline cognition from frailty—more than chronological age—influences adverse-event risk; judicious selection and complication-sensitive technique can provide meaningful functional outcomes post-metastatic surgery despite old age [33]. Of interest, in the consensus delirium statements amended in 2025, the focus is on multimodal, non-pharmacologic prevention of delirium, except for antipsychotic medications at cautionary low dosages, for an exceptionally brief duration, only in cases in which agitation is unpleasant and threatens care [34]. The brief psychiatric episode in this patient in the backdrop of her neurodegenerative Alzheimer’s disease, and the resumption of a ‘normal’ state as evidenced by low-dose haloperidol medication and normalization of sleep and rehabilitation, would comfortably fall within those recommendations.

### 3.6. Venous Infarction, Hydrocephalus, and Posterior Fossa Specific Technical Challenges

Posterior fossa resections are much more vulnerable regarding the protection of venous drainage (superior petrosal complex, hemispheric bridging veins) and preservation of foramina of Luschka and Magendie and to avoid obstructive hydrocephalus. Contemporaneous operative series and atlases demonstrate that systematic arachnoid dissection in the context of venous veils, gravity-dependent/surgical vectors, and subpial corticotomies on the axis of folia can minimize the rates of venous infarction and pseudomeningoceles—as demonstrated here [35].

### 3.7. Radiotherapy and Cognition in Patients with Pre-Existing Neurocognitive Disorders

In patients with dementia, regarding time frame as a WBRT candidate, hippocampal-sparing techniques and the addition memantine are becoming important, as phase-III data suggest progressively preserved cognition and patient-reported outcomes for HA-WBRT + memantine as compared to WBRT alone [36,37]. Several upcoming plans in 2024–2025 are to optimize dose scheduling and also to extend “memory-avoidance” sparing outside of the hippocampi in patients who have many intracranial lesions. If distant intracranial disease control were to ever require WBRT eventually in their disease, these paths offer real options [38].

### 3.8. Economics, Access, and Patterns of Practice

From a health-systems perspective (recent studies that incorporated national payer recommendations), practice models based on SRS modalities, as opposed to WBRT-heavy approaches for managed care, can deliver cost savings at least intermittently, at the same time returning outcomes. Even with SRS influences factored in relative to the type of treatment, due to the near daily number of new targeted agents that receive code discovery and use in practice, concerning aspects of failed-system costs exist across inpatient, outpatient, and pharmacy systems [39]. National registry research in 2024 showed the increasing use of SRS relative to use of WBRT for limited numbers of metastasis, which is a concern, as the inequity of access is an evident issue that will have to be considered in future service planning [40].

### 3.9. How This Case May Inform Clinical Practice

We can identify some practical points. With the rapid decompression of symptomatic posterior fossa disease, we may be able to reinstate some degree of function, even in very old adults. No neuroimaging modality can definitively be considered more or less accurate than other determinants, only factors—bonafide resectable planes and of course cerebral spinal fluid pathways that require restoration. The very early point at which the fourth ventricle was patent (CT and non-distortion of brainstem for immediate post-operative imaging) offered the ‘right’ conditions for rehabilitation, and potential adjuvant planning.

Secondly, being complete with the diagnosis matters—an IHC-driven work-up (and molecular profiling when relevant) may change the direction of management away from managed care surgery and radiation to CNS-active systemic therapy based on lung-cancer characterizations or others with substantial CNS disease activity, if they are characterized.

Thirdly, sequencing for radiation is not prescriptive and should be managed on an individual basis; the current literature supports if you are using SRS for cavity treatment after resection, with active trials and prospective cohorts delineating the impact of pre-operative SRS + LMD and toxins, especially within a posterior fossa disease context.

Finally, geriatric and cognitive comorbidities are not barriers to functional recovery provided there is prurient planning during the peri-operative phase and survivorship to manage delirium prevention and cognitive-sparing radiation delivery methods when WBRT is a possibility.

## 4. Conclusions

The purpose of this case is to show how a systematic neurological examination, intensive neuroimaging assessment, and microsurgical dissection with anatomical specificity can safely treat posterior fossa lesions. In this case, clinical localization and imaginative planning were made to develop an approach that would lessen the mass effect and respect venous drainage patterns and eloquent neural tracts. Although this report is limited to one patient’s experience, we hope to provide this case to help contribute to knowledge of decision-making in complex surgery of the posterior fossa. The approach described is based on principles that are standard in practice, assessing risk of the individual, protecting functional reserve, and minimizing morbidity, alongside recognizing that advanced imaging, molecular characterization, and the ideal sequencing of adjuvant therapy and interventions evolve over time. In the future, it is possible that we could consider incorporating higher resolution tractography for variable resolution afferent mapping of cerebellar networks, using perfusion-based modeling of risk to predict expected post-operative changes, and incorporating, earlier in the planning phase, molecular characteristics to depict tumor profiles. Future studies continue to investigate the best sequencing of surgery, radiotherapy, and systemic therapy in posterior fossa disease; it is hoped that this will help clarify patient selection processes and optimize functional outcomes.

By providing a comprehensive description of the patient’s presentation, operative course, and early recovery, we hope that this case report provides a useful reference for treating clinicians dealing with similar cases, bearing in mind that every case will be managed differently in different clinical and anatomical contexts.

## Figures and Tables

**Figure 1 diagnostics-15-03131-f001:**
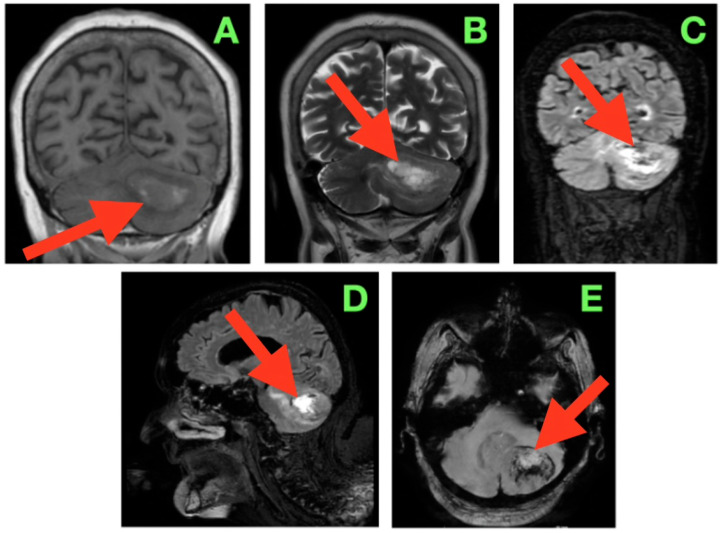
Pre-operative magnetic resonance imaging of the brain demonstrating a solitary left cerebellar hemisphere mass with associated vasogenic edema and mass effect on adjacent infratentorial structures. The lesion is approximately 40 mm in diameter as measured by multi-planar MRI in each axis (AP ≈ 40 mm; ML ≈ 40 mm; CC ≈ 40 mm) and thus has a calculated volume of about 30–35 cm^3^ based on the ABC/2 formula. In addition, the upper margin of the lesion is less than 3–4 mm from the tentorial insertion, and the lower margin of the lesion is greater than 8–10 mm from the foramen magnum with preservation of the cisterna magna. The lesion contacts but does not invade the dentate nucleus and displaces the fibers of the middle cerebellar peduncle. (**A**) Coronal T1-weighted image showing an iso- to mildly hypointense lesion (arrow) expanding the left cerebellar hemisphere, with the superior margin following the tentorial surface, medial indentation of the superior vermis, and rightward displacement of the fourth ventricle without overt hydrocephalus. (**B**) Coronal T2-weighted image depicting a hyperintense lesion (arrow) with internal signal heterogeneity and surrounding vasogenic edema radiating toward the middle cerebellar peduncle; the edema distribution involves the trajectory of dentatothalamic and corticopontocerebellar pathways. (**C**) Coronal FLAIR sequence in which suppression of CSF signal enhances delineation of the edema margins (arrow), showing medial extension into the paravermian white matter, narrowing of the perivermian cistern, and gentle anterior displacement of the dorsal pons. (**D**) Sagittal FLAIR view demonstrating the anteroposterior extent of the lesion (arrow) from the region anterior to the posterior medullary velum to the convexity of the cerebellar hemisphere; the superior margin lies in close apposition to the tentorial leaflet, while the inferior border remains above the foramen magnum, preserving the cisterna magna. (**E**) Axial susceptibility-weighted image revealing multiple punctate hypointense foci within the lesion (arrow), suggestive of intralesional blood products, with a sharply demarcated border from surrounding edema and no evidence of additional posterior fossa lesions.

**Figure 2 diagnostics-15-03131-f002:**
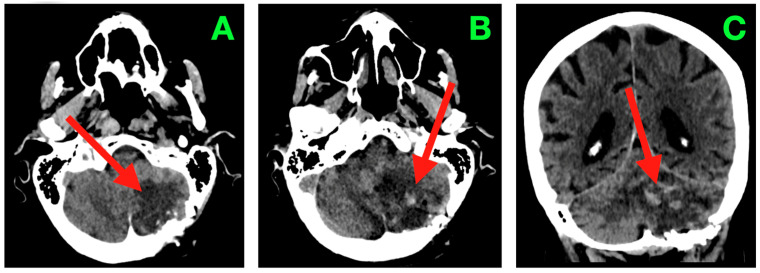
Immediate post-operative non-contrast CT scan. (**A**) Axial section through the inferior cerebellar convexity showing the post-operative resection cavity occupying the left neocerebellar hemisphere (arrow), with smooth posterolateral margins abutting the suboccipital craniectomy window. The inferior wall terminates above the foramen magnum, and no residual mass effect on the cervicomedullary junction is present. (**B**) Axial section at the level of the superior cerebellar hemisphere and tentorial undersurface demonstrating the superior margin of the cavity (arrow) in close apposition to the tentorial surface, with preserved bridging veins and absence of hyperdense foci indicative of venous thrombosis or parenchymal hemorrhage. (**C**) Axial section through the fourth ventricle depicting a midline vermis contour preserved in its entirety, with the cavity’s medial border (arrow) respecting the paravermian zone. The outlets of the fourth ventricle (foramina of Luschka and Magendie) are patent, and CSF freely communicates with the cisterna magna. No evidence of intraventricular hemorrhage, hydrocephalus, or brainstem distortion is observed.

**Figure 3 diagnostics-15-03131-f003:**
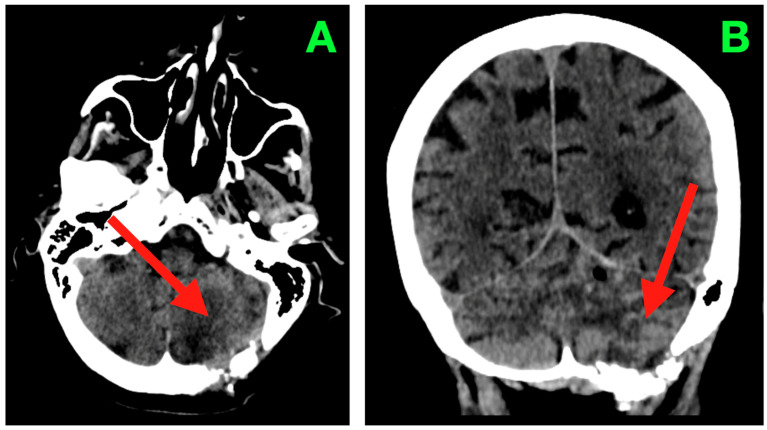
Two-week post-operative CT scan. (**A**) Axial non-contrast CT through the posterior fossa demonstrates a sharply delineated post-operative resection cavity within the left neocerebellar hemisphere (arrow), corresponding to the site of tumor excision. The cavity margins are smooth and gliotic, without irregularity or nodularity to suggest residual tumor. The surrounding cerebellar parenchyma exhibits homogeneous attenuation with no focal hypoattenuation suggestive of ischemia and no hyperdense collections indicating hemorrhage. The ipsilateral transverse and sigmoid sinuses remain patent, and the fourth ventricle has returned to midline with restored anteroposterior diameter. Perifocal edema present in the immediate post-operative period has resolved, as evidenced by normalization of parenchymal density and the absence of sulcal effacement. The basal cisterns are widely patent, including the prepontine, cerebellopontine, and cerebellomedullary cisterns, confirming unobstructed CSF pathways in the posterior fossa. (**B**) Coronal CT reconstruction depicts the vertical profile of the resection cavity (arrow), extending from the superior folial tiers of the cerebellar hemisphere to just above the foramen magnum, while preserving the integrity of the vermis and sparing the dentate nucleus medially. The tentorial undersurface maintains its normal concavity, with no evidence of superior displacement or distortion. The cisterna magna appears well-formed and free of mass effect. No pneumocephalus, extra-axial collections, or pseudomeningocele are present. These findings are consistent with a complete macroscopic resection, preservation of critical cerebellar and brainstem structures, and a complication-free post-operative evolution at this time point.

**Figure 4 diagnostics-15-03131-f004:**
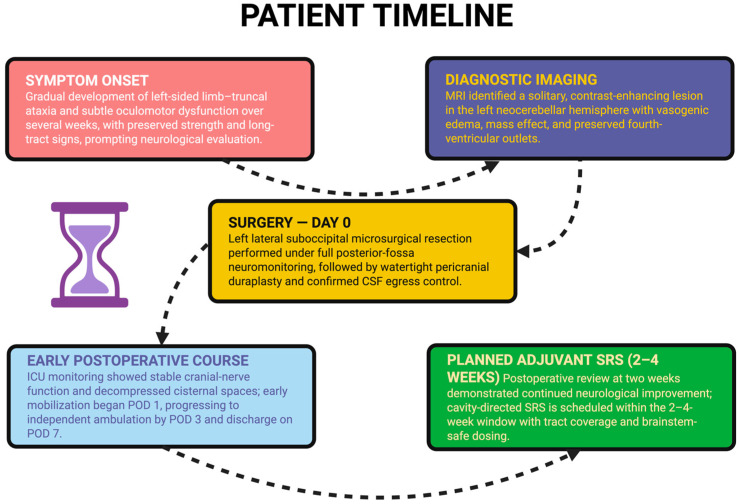
A chronological summary of the patient’s course from symptom onset to planned adjuvant therapy, including initial neurological decline, diagnostic MRI, microsurgical resection, early post-operative recovery, and scheduling of cavity-directed stereotactic radiosurgery within the 2–4-week window.

**Table 1 diagnostics-15-03131-t001:** Summary of selected original studies (2020–2025) reporting outcomes and management strategies for posterior fossa brain metastases. Data include study design, patient demographics, primary cancer distribution, treatment modalities, adjuvant therapies, survival outcomes, complication rates, and key novel findings. Only studies with a substantive proportion of infratentorial lesions were included to ensure direct clinical relevance.

References	Design/Cohort	Key Population	Therapy	Outcomes	Practice-Relevant Notes
[6]	Retrospective surgical cohort	73 cerebellar mets; lung/breast/GI common	Microsurgery; hydrocephalus relief	Median OS 9.2 mo; hydrocephalus worsened survival	Posterior fossa surgery yields outcomes comparable to supratentorial disease when selected carefully
[16]	Retrospective surgical series	57 cerebellar mets; NSCLC/CRC/breast	Resection; multimodal care	IC-PFS 4.5 mo; OS 11.6 mo	Underscores early planning for cavity SRS given infratentorial recurrence risk
[17]	Comparative cohort	Elderly cerebellar mets	Tailored microsurgery	Functional recovery achievable	Frailty > age in determining safety—supports surgery in older adults
[18]	Prospective feasibility (pre-op SRS)	Resectable mets incl. posterior fossa	Pre-op SRS 24–72 h before surgery	Feasible; early cavity control; exploratory LMD signal	Concept of sterilizing tumor–CSF interface informs sequencing debate
[19]	Clinical series (post-op SRT)	Post-resection cavities	Fractionated SRT (e.g., 24 Gy/3 fx)	High LC; low brainstem toxicity	Fractionation mitigates posterior fossa dose-constraint limitations
[20]	Real-world palliative cohort	Predominant posterior fossa disease	Whole posterior fossa RT	Symptom relief	Useful when multifocal PF disease precludes focal SRS
[21]	Institutional posterior fossa experience	Post-op cavities	PORT/cavity RT	Higher LR/LMD with large volumes or pseudomeningocele	Highlights importance of cavity geometry and timing for SRS
[22]	Feasibility (delayed resection)	Resectable mets	Pre-op SRS with delayed surgery	Safe; immunologic interest	Extends pre-op SRS concept where PF logistics delay immediate surgery
[23]	Contemporary SRS practice update	Deep/large cavities	Cavity SRS with tract inclusion	LC tempered by dose-constraint adaptations	Directly relevant: brainstem constraints often require fractionation + tract coverage

## Data Availability

The raw data supporting the conclusions of this article will be made available by the authors on request.

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
