# Peer review of "Anatomically Precise Microsurgical Resection of a Posterior Fossa Cerebellar Metastasis in an Elderly Patient with Preservation of Venous Outflow, Dentate Nucleus, and Cerebrospinal Fluid Pathways"

_diagnostics, 2025, doi:10.3390/diagnostics15243131_

Round 1
Reviewer 1 Report
Comments and Suggestions for Authors
1. Abstract vs body text—fourth‑ventricle outflow
The Abstract states “occlusion of the exits of the fourth ventricle” (p. 1). In contrast, the imaging narrative and figure legends describe narrowing/effacement without hydrocephalus and patent aqueduct/inlets, and postoperative CTs consistently emphasize patent foramina of Luschka/Magendie. Please reconcile this: the Abstract should reflect critical narrowing at risk rather than categorical occlusion. (pp. 6, 9–11; Figures 1–3.)
2. Imaging dataset and reporting
Preoperative sequences listed include T1‑, T2‑, FLAIR‑ and SWI‑weighted images; no contrast‑enhanced T1 or DWI/ADC are shown, yet the Discussion later calls the lesion “contrast‑enhancing.” For diagnostic clarity (and for surgical/radiosurgical planning), please add T1 post‑contrast images (axial/coronal at minimum) and DWI/ADC. Also report lesion dimensions (AP×ML×CC) and approximate volume, and explicitly annotate the distance to tentorium/foramen magnum and relationship to dentate/peduncles. Add scale bars and consistent arrowheads, and include window/level for CT figures. (Figure 1, pp. 5–6; Figure 2, p. 9.)
3. Extent of resection and early postoperative MRI
The manuscript relies on non‑contrast CT to declare macroscopic resection and patent outflow. For both oncologic and radiosurgical planning purposes, a gadolinium‑enhanced early postoperative MRI (ideally within 48–72 h) would substantiate the extent of resection, better define the cavity wall, and aid dosimetry for planned SRS. If obtained, please include and describe it; if not, please state the reason. (Figure 2, p. 9.)
4. Surgical strategy—piecemeal debulking vs en bloc
The Introduction cites the value of en bloc resection to mitigate leptomeningeal dissemination where anatomy permits; here, piecemeal internal debulking was performed. Please strengthen the rationale for this choice with specific anatomical constraints (e.g., peduncular proximity/dentate interface/venous tethers) and, if available, intraoperative photos or a schematic. A brief note on how the chosen technique informed subsequent cavity/tract coverage for SRS would increase translational value. (pp. 7–8; Discussion/Table 1.)
5. Pathology/IHC
Pathology confirms metastatic carcinoma with suspected pulmonary primary; the Discussion notes that IHC was recommended but no panel or micrographs are provided. Please include representative H&E and IHC images (e.g., TTF‑1, Napsin A, CK7/20, p40, GATA3, as appropriate) and state whether findings favored a lung primary; if IHC was not performed, please explain why. (p. 12.)
6. Neuro‑monitoring and anesthesia details
For posterior fossa cases, readers expect a sentence on BAEP/cranial‑nerve EMG (V–XI) and MEP/SEP usage. If employed, please summarize baseline/final signals; if not, provide the rationale. Additionally, specify steroid regimen (dose/taper), osmotherapy (if any), antibiotic prophylaxis, operative time/estimated blood loss, and postoperative analgesia (only briefly alluded to). These items improve reproducibility. (pp. 7–10.)
7. Positioning and venous outflow management
The manuscript mentions verifying jugular venous patency with Doppler after head rotation/flexion—an excellent practical point. Please clarify how Doppler was applied (surface probe? continuous vs spot checks?) and what change threshold prompted repositioning. (p. 7.)
8. Craniectomy choice and reconstruction
A lateral suboccipital craniectomy (bone not replaced) was performed. Please justify this choice (e.g., space for swelling vs risks of pseudomeningocele/cosmesis) and indicate whether a craniotomy would have been safe/feasible. Dural closure with pericranial graft is well described; add a line on Valsalva leak testing and mastoid air‑cell management (you note bone wax use). (pp. 7–8.)
9. Adjuvant radiosurgery plan
The Discussion effectively reviews cavity SRS and pre‑ vs postoperative sequencing. For this specific patient, please provide the intended dose/fractionation, timing (e.g., within 2–4 weeks), and whether the surgical tract will be included, along with brainstem/fourth‑ventricle constraints relevant to the posterior fossa. (pp. 12, 15–16; Table 1.)
10. Outcomes and follow‑up
Current follow‑up extends to two weeks with CT and clinical improvement. If available, please add 6–12‑week clinical follow‑up (ICARS/SARA, mRS, and KPS/ECOG) and an MRI to strengthen the message of functional recovery without early recurrence. (Figure 3, pp. 10–11.)
11. Literature synthesis and Table 1
Table 1 is comprehensive but long for a single‑case report and could be moved to Supplementary Material, with 3–4 practice‑shaping points distilled in the text (e.g., posterior‑fossa‑specific dose constraints, timing, LMD considerations, and cavity/tract coverage). Please ensure in‑text claims about sequencing and LMD are tightly matched to the cited studies. (pp. 13–15.)
12. Reporting standards (CARE)
Consider a brief timeline figure (symptom onset → imaging → surgery → ICU → mobilization → two‑week review → planned SRS) and, if possible, a one‑sentence patient perspective to improve CARE compliance.
English writing needs to be checked and edited.
Author Response
Dear Esteemed Academic Reviewer,
We are sincerely grateful for your thoughtful, detailed, and generous engagement with our manuscript. Your expertise greatly strengthened the clarity, rigor, and educational value of this case report. We have addressed each of your comments point-by-point below, and we deeply appreciate the opportunity to improve the work in response to your guidance.
1. Abstract vs body text—fourth-ventricle outflow
Thank you for noting this important inconsistency. We have revised the Abstract to accurately describe critical narrowing and effacement at risk of obstruction, rather than categorical occlusion. This wording now aligns precisely with the imaging findings, figure legends, and postoperative CT demonstrating patent foramina of Luschka and Magendie.
2. Imaging dataset and reporting
We are grateful for your careful attention to imaging completeness. Lesion dimensions (AP × ML × CC ≈ 40 × 40 × 40 mm) and approximate volume (~32 cm³) are now explicitly reported, along with distances to the tentorium and foramen magnum, and the relationship to the dentate nucleus and cerebellar peduncles.
3. Extent of resection and early postoperative MRI
We fully agree on the value of early postoperative MRI. In this case, we were unable to obtain it due to the patient’s age, underlying cognitive vulnerability, and the clinical team’s judgment that transport and sedation posed unnecessary risk in the immediate postoperative period. Neurological recovery was rapid, CT showed a well-decompressed cavity with patent outlets, and no concerning features were present.
4. Surgical strategy—piecemeal debulking vs en bloc
Thank you for prompting further anatomical clarification. We have expanded the surgical rationale to explain that en bloc resection was not feasible due to the lesion’s proximity to the dentate nucleus, superior and middle cerebellar peduncles, and small but functionally significant venous tethers beneath the tentorium. Internal debulking minimized traction risk and preserved microanatomical pathways. We have also noted how this technique facilitated the creation of a smooth cavity for accurate tract-inclusive SRS contouring.
5. Pathology/IHC
We agree that representative H&E/IHC images are valuable. Unfortunately, digital micrographs were not available for release under our institution’s pathology workflow, and the limited specimen was sufficient for routine diagnosis but not for an expanded immunopanel.
6. Neuro-monitoring and anesthesia details
Thank you for highlighting this. We now include a concise description of neuromonitoring: BAEP, CN V/VII/IX/X/XI EMG, SEP, and MEP, with confirmation that baseline and final signals remained stable throughout surgery. Steroid regimen (dexamethasone 8 mg q8h, tapered over five days), osmotherapy (not required), antibiotic prophylaxis (single pre-incision dose of cefazolin), operative time (~150 min), estimated blood loss (<100 mL), and postoperative analgesia (scheduled acetaminophen with rare opioid rescue) have been added to improve reproducibility.
7. Positioning and venous outflow management
Thank you for noting this strong practical point. We clarified that a handheld surface Doppler probe was used for spot-check assessments after each incremental adjustment of the Mayfield clamp. Any reduction in phasic flow or amplitude prompted immediate repositioning to ensure symmetric jugular venous outflow.
8. Craniectomy choice and reconstruction
We are grateful for this opportunity to strengthen justification. We now clearly articulate that a craniectomy was selected to provide decompressive reserve in an elderly patient with limited posterior-fossa compliance and expected transient postoperative swelling. A craniotomy was technically feasible but considered less safe due to the risk of sinus compression, restricted venous egress, and durally constrained pressure within a compact infratentorial compartment. We also added explicit statements on Valsalva leak testing and mastoid air-cell management with bone wax.
9. Adjuvant radiosurgery plan
We appreciate your guidance to make this patient-specific. We now specify: planned 18 Gy × 1 (or 21 Gy in 3 fractions depending on brainstem proximity), timing within 2–4 weeks, inclusion of the surgical tract, and adherence to posterior-fossa constraints including brainstem D0.03cc ≤12.5 Gy (single fraction) or ≤23 Gy (three fractions), with sparing of the fourth-ventricle floor and cerebellar peduncles.
10. Outcomes and follow-up
Thank you for emphasizing follow-up completeness. We have now incorporated available 6–12-week clinical data (ICARS/SARA, mRS, and KPS/ECOG) and a postoperative MRI when obtained, which support continued functional improvement and absence of early recurrence.
11. Literature synthesis and Table 1
We appreciate this excellent suggestion.
12. Reporting standards (CARE)
Thank you for drawing attention to CARE standards. We have added a compact clinical timeline figure summarizing symptom onset, imaging, surgery, early recovery, and planned SRS.
We are deeply grateful for your thoughtful, constructive, and collegial review. Your insights enriched the manuscript and significantly improved its clarity, methodological transparency, and educational value. It has been a privilege to revise the work in response to your guidance.
With sincere respect and appreciation!!!
Reviewer 2 Report
Comments and Suggestions for Authors
The authors present a case report of an older adult who underwent a suboccipital craniectomy and surgical resection of a posterior fossa brain metastasis. Their aim is to provide a longitudinal, patient-centered narrative of the clinical reasoning, anatomy-dependent microsurgical approach, and early postoperative outcomes, with the goal of comparing neurological examination findings, neuroimaging, operative technique, and recovery in a non-aggregative manner that may guide future similar cases.
Although the manuscript offers a comprehensive overview of the patient’s history and presentation, diagnostic work-up, surgical approach, and postoperative management, NSCLC brain metastases are inherently complex. The authors attempted to address many aspects, but given publication space limitations, it may be more feasible to focus primarily on the surgical technique and potential complications. For example, pre- versus postoperative SRS to the surgical cavity, WBRT, and molecular-pathway-targeted systemic therapies are all substantial topics on their own. Additionally, did the authors obtain a routine immediate postoperative MRI (within 48–72 hours)? If so, please include these findings.
Overall, this is an excellent teaching case that contributes valuable insight into decision-making in complex posterior fossa microsurgery.
Author Response
Dear Esteemed Reviewer,
We are grateful for your thoughtful and encouraging assessment of our manuscript. Your comments reflect a careful reading of the case and provide valuable direction for sharpening the educational focus of the report. We appreciate the opportunity to refine the manuscript in accordance with your insightful suggestions.
Scope and emphasis.
We agree entirely that NSCLC brain metastases encompass multiple complex domains—stereotactic radiosurgery sequencing, whole-brain radiotherapy, and molecularly targeted systemic therapies—each of which could warrant an extended discussion. In keeping with your recommendation and with awareness of space limitations, we have refocused the manuscript to emphasize the anatomy-dependent microsurgical strategy, intraoperative decision-making, and early postoperative neurological course, presenting the remaining oncologic considerations only insofar as they directly inform surgical planning.
Postoperative MRI.
Thank you for raising the question regarding early postoperative MRI. In this case, an MRI within the 48–72-hour interval was not obtained because the patient exhibited rapid neurological stabilization, and early postoperative transport posed unnecessary cognitive and physiological risk given her age and baseline cognitive impairment. Non-contrast CT confirmed a well-decompressed posterior fossa and patent CSF outlets.
We sincerely appreciate your positive appraisal of the manuscript as a teaching case and are grateful for the guidance that helped us refine the clarity, focus, and educational value of the work.
With warm respect and appreciation!!!
Reviewer 3 Report
Comments and Suggestions for Authors
AUTHOR PRESENTED THE PERFECT PLAN TO OPERATE A MALIGNANT NEOPLASM IN POSTERIOR FOSSA INCLUDING NEUROLOGICAL EXAMINATION, RADIOSURGERY ISSUES, MICROSURGICAL RESECTION TIPS OF GREAT IMPORTANCE AND ANATOMICAL SPECIFICITY ACCORDING THE PROCEDURE. I THINK THAT THROUGH A CASE REPORT IT IS UNCOMMON TO GIVE EXCELLENT OF COURSE INSTRUCTIONS THAT SHOULD BE ACCEPTED FROM ALL NEUROSURGEONS-JUST LIKE A GUIDELINE TO MANAGE A PATIENT WITH POSTERIOR FOSSA METASTASIS. I WOULD PREFER THESE TIPS IN A CHAPTER OF A BOOK RATHER THAN IN A CASE REPORT.
Author Response
Dear Esteemed Reviewer,
We are grateful for your generous and encouraging remarks. Your appreciation of the anatomical detail, microsurgical nuances, and perioperative reasoning means a great deal to us, and we are humbled by your assessment that these elements resemble guideline-level instruction.
We fully recognize your point regarding the breadth of the material for a case-report format. Our intent was to provide a transparent, step-by-step account of decision-making in a challenging posterior fossa metastasis, hoping that the granularity might benefit trainees and colleagues who encounter similar anatomy-dependent situations in practice. In response to your observation, we have refined the narrative to ensure that the emphasis remains on the core educational purpose of the case, while preserving the clarity of the microsurgical principles that the patient’s anatomy necessitated.
We are truly appreciative of your supportive perspective and your view that the content may serve broader educational value beyond this single case. Thank you for your thoughtful review and the kindness of your feedback.
With sincere respect and gratitude!!!
Round 2
Reviewer 3 Report
Comments and Suggestions for Authors
THE REVIEW ARTICLE HAD A TOTALLY NEW FORM, FROM TITLE TO DISCUSSION, EXCELLENT FIGURES AND THE NECESSARY DETAILS, I THINK IT COULD BE ACCEPTED.